# Etiologic Classification of Diffuse Parenchymal (Interstitial) Lung Diseases

**DOI:** 10.3390/jcm11061747

**Published:** 2022-03-21

**Authors:** Matthias Griese

**Affiliations:** Department of Pediatric Pneumology, Dr. von Haunersches Kinderspital, University of Munich, German Center for Lung Research, Lindwurmstr. 4a, D-80337 Munich, Germany; matthias.griese@med.uni-muenchen.de

**Keywords:** children’s interstitial lung disease (chILD), human phenotype ontology, interstitial pneumonia, interstitial pneumonitis, idiopathic interstitial fibrosis, familial, surfactant, classification, categorization

## Abstract

Interstitial lung diseases (ILD) or diffuse parenchymal lung diseases (DPLD) comprise a large number of disorders. Disease definition and classification allow advanced and personalized judgements on clinical disease, risks for genetic or environmental transmissions, and precision medicine treatments. Registers collect specific rare entities and use ontologies for a precise description of complex phenotypes. Here we present a brief history of ILD classification systems from adult and pediatric pneumology. We center on an etiologic classification, with four main categories: lung-only (native parenchymal) disorders, systemic disease-related disorders, exposure-related disorders, and vascular disorders. Splitting diseases into molecularly defined entities is key for precision medicine and the identification of novel entities. Lumping diseases targeted by similar diagnostic or therapeutic principles is key for clinical practice and register work, as our experience with the European children’s ILD register (chILD-EU) demonstrates. The etiologic classification favored combines pediatric and adult lung diseases in a single system and considers genomics and other -omics as central steps towards the solution of “idiopathic” lung diseases. Future tasks focus on a systems’ medicine approach integrating all data and bringing precision medicine closer to the patients.

## 1. Introduction

Disease definition and classification play an important role for progress in human health. The taxonomies used directly reflect the underlying principles for best diagnosis, treatment, and prophylaxis. The more precise the underlying complexities are studied, and this knowledge links to the disease of the individual, the better judgements on disease course, prognosis, and risks for genetic or environmental transmissions are targeted in personalized precision medicine.

A few decades ago, drugs with similar actions, e.g., different ß-blockers or inhaled steroids with only minor modifications, for usage in common diseases such as hypertension, asthma, and COPD dominated the progress in medicine. Currently, exploitation of molecularly well-defined, mostly rare diseases is in the center of rapid developments of novel, targeted treatments. Detailed pathway knowledge and hypothesis-free approaches make so far only symptomatically treatable conditions now drugable. An example is the small molecule medications used in cystic fibrosis, making a substantial correction of the underlying genetic CFTR defect possible [1,2].

Splitting diseases into smaller and smaller entities is key for precision medicine. Nevertheless, the smart merging of etiologically similar diseases into categories (lumping) allows targeting of similar diagnostic or therapeutic principles and more convenient memorizing in clinical practice. Such developments occur in subspecialty niches at a rapid pace. Specialized registers nicely capture the rich spectrum of individuals with their disease and can then build growing cohorts of well-defined novel entities. At the same time, such registers make the disease collections open to computerized ontologies and integration of translational multi-omics and precision medicine.

Here, we summarize the development and practice of categorization systems in the area of diffuse lung diseases from the experiences made in the children’s lung register/chILD-EU register, a European pediatric rare lung disease register and biobank [3]. In the context of current etiologic disease categorization, we discuss some issues of everyday interest and set the stage for future developments for the integration of physicians’ and systems’ biology scientists’ needs in an age-overarching approach.

## 2. Terminology—DLD, DPLD, ILD, chILD

*Diffuse lung disease* (DLD) is a term infrequently used. DLDs are lung diseases that involve the entire lungs, i.e., from base to top, peripheral to central, and front to back (Appendix A, Table A1). They can present in a patchy way or uniformly. Many airway diseases such as asthma and COPD, or rarer ones such as cystic fibrosis and primary ciliary dyskinesia are DLD, whereas neoplasms, pleural disorders, or gross structural lung abnormalities affect the lungs much less frequently diffusely (Figure 1). Here, we focus on the large group of *diffuse parenchymal lung diseases* (DPLDs) or *interstitial lung diseases* (ILDs). Parenchyma is the functional part of an organ; in the lungs it comprises the components that are involved in gas exchange. This includes the respiratory bronchioles, the pulmonary alveoli, and the extracellular matrix and cells, outside of the lungs’ circulatory system. The latter distinction is the reason why vascular disorders are sometimes departed from parenchymal disorders. Here, we include them into the group of ILDs. Lung interstitium is a network of connective tissue that supports the structure of the organ. We and others elsewhere [4] do not distinguish between diseases that affect the lungs’ interstitium or its parenchyma, but use the terms as synonyms.

Damage to lung parenchyma by varying patterns of inflammation and fibrosis results in this large and heterogeneous group of non-neoplastic disorders. Importantly, these disorders frequently affect not only the parenchyma but also the peripheral airways and vessels along with their epithelial and endothelial linings. This is the reason why some airway diseases, such as obliterating bronchiolitis and the vascular diseases, are listed under ILD.

ILD in children or pediatric ILD has been collected under the acronym chILD (or chILD syndrome) in order to memorize easily and to identify a phenotype that requires prompt and expert evaluation [5]. This separation has been helpful in the past to highlight an orphaned field of diseases. However, it has become clear that such diseases may also occur in adults. If reserved for a certain age group, these diseases may not be diagnosed appropriately, when not expected to present at other ages. Thus, we prefer an age-overarching approach.

First, we focus on the classification systems for interstitial lung diseases, giving a brief history. Then, we emphasize the value of etiologic classification also for registers, and present the system currently applied therein. Lastly, we look on some future needs of registers to realize personalized medicine for ILDs.

## 3. Brief History of ILD Classification Systems

In adults, where the frequency of ILD is at least more than 10-fold larger than in children [6], many ILDs are diagnosed unambiguously [7]. These include sarcoidosis, hypersensitivity pneumonitis, or Langerhans histiocytosis; such conditions were traditionally marginal for the intense work done in adult pulmonology to differentiate the idiopathic interstitial pneumonias [7]. In contrast, the pediatric classifications covered all the entities just mentioned due to their rarity in children, introduced novel, molecularly defined ILDs, such as the surfactant dysfunction disorders, and included new disease categories linked to the disruption of lung development. In the following section, we review and align the entities of the major published ILD classification systems used in adults and children. Our goal was to provide a harmonized view, which allows an age-overarching approach that is useful for the integration of clinical, register, and molecular work (Table 1).

### 3.1. The Idiopathic Interstitial Pneumonia (IIP) Classification Systems in Adults

In 1969, Liebow and Carrington proposed a classification for the chronic forms of interstitial pneumonias. This and following categorizations until today were mainly based upon histopathology. The authors differentiated five types: usual interstitial pneumonia (UIP), bronchiolitis obliterans interstitial pneumonia and diffuse alveolar damage (BIP), desquamative interstitial pneumonia (DIP), lymphocytic interstitial pneumonia (LIP), and giant cell interstitial pneumonia (GIP) [8]. In that classification the different histopatterns could be caused by microbial or toxic exposures; they did not focus on the idiopathic forms. During the following decades, the concept of “idiopathic”, i.e., those without known etiology, disease groups developed further [9]. In particular, Katzenstein proposed several revisions and updates, the latest classification including four forms of idiopathic interstitial pneumonias, i.e., UIP, DIP/respiratory bronchiolitis interstitial lung disease, acute interstitial pneumonia, and non-specific interstitial pneumonia [10]. Müller and Colby added bronchiolitis obliterans organizing pneumonia to that collection [11].

In 2002, the American Thoracic Society/European Respiratory Society International Multidisciplinary Consensus on the Classification of the Idiopathic Interstitial Pneumonias [7] was published to set an international standard, which was a first move away from the primacy of pathology in categorizing the idiopathic interstitial pneumonias (IIP). This was necessary with the advent of high-resolution computerized tomography and the focus on a multidisciplinary team approach, trying to include also patients who did not have a diagnosis based on a lung biopsy for various reasons. The goal of the consensus was to better define the clinical manifestations, the histo-pathology, and the radiologic features of patients with IIP. Seven forms of IIPs were differentiated (Table 1). In their publication, they further differentiated two groups of DPLD with known cause, a group with granulomatous DPLD, and other forms of DPLD. A major paradigmatic change of this new classification was the definition of a set of histologic patterns, which provided the basis for a final clinico-radiologic-pathologic diagnosis. This multidisciplinary team approach still put the primary weight on the pathologist because histologic pattern result was rated over the imaging patterns seen by radiologists. However, the final clinic-pathologic diagnosis, including a judgement of the disease as “idiopathic”, should only be made after an in-depth review of all clinical, imaging, and histological data by the pulmonologist, radiologist, and pathologist.

In 2013, a revision of the 2002 IIP classification was published, integrating important developments and experiences from the previous decade [12]. The seven main disease entities were preserved. Among these, idiopathic NSIP was accepted as a distinct clinical entity and no longer as “provisional”. Some important disease characteristics were more stressed by extra-grouping, i.e., chronic fibrosing IIPs (IPF, NSIP), smoking-related conditions (RB-ILD, DIP), and acute/subacute IIPs (cryptogenic organizing pneumonia (COP) and acute interstitial pneumonia (AIP)). The rare histologic patterns of acute fibrinous and organizing pneumonia (AFOP) and interstitial pneumonias with a bronchiole-centric distribution were recognized. Idiopathic pleuro-parenchymal fibroelastosis and unclassifiable idiopathic interstitial pneumonias were added as new groups. The latter was an important, clinically relevant step. In addition, for the first time, grading of disease behavior was included. This acknowledged the heterogeneity of the differential natural progression of IIPs, i.e., of NSIP and IPF. In the same line, the role of acute exacerbations and of molecular and genetic disease markers was put into perspective.

In 2015, an official European Respiratory Society/American Thoracic Society research statement introduced the quite controversial disease term of interstitial pneumonia with autoimmune features (IPAF) [13]. IPAF should be used to identify individuals with IIP and features suggestive of, but not definitive for, a connective tissue disease (CTD).

### 3.2. The Pediatric Classification Systems (chILD)

In 2004, a report of the “European Respiratory Society Task force on chronic interstitial lung disease in immunocompetent children” stated that these disorders were poorly investigated in children and most of the strategies applied by pediatricians were linked to studies in adult patients [14]. Basically, the categorization scheme presented was similar to that of the 2002 adult statement, except that “congenital disorders” were added as a new group and several other important pediatric entities were listed under “Other forms of interstitial pneumonia”. To the latter belonged alveolar proteinosis, eosinophilic pneumonia, infantile/idiopathic hemosiderosis, persistent tachypnea of infancy, and pulmonary interstitial glycogenosis. Although very heterogeneous, these entities made up a decent fraction of ILD in children. Of importance, genetically defined surfactant dysfunction disorders (SP-B, SP-C deficiency), forms of familial cryptogenic fibrosing alveolitis, and molecularly well-defined syndromes such as Hermansky–Pudlak syndrome or IPPs due to inborn errors of metabolism were added. This was a remarkable advance, going beyond the adult classification systems.

In 2007, a mainly pathology-based classification scheme for diffuse lung disease in children under the age of 2 years was published [15]. It was based on the analysis of 187 cases with lung biopsy. The most prominent feature was the clearer separation of four categories of disorders more prominent in infancy from the other disorders (Table 1). Among these disease categories were disorders of early lung development. If they affected the entire lungs, they were called diffuse developmental disorders [15]. Such disorders lead to severe structural and functional lung abnormalities and are often lethal, with typical examples being alveolar capillary dysplasia with misalignment of the pulmonary veins, acinar dysplasia, and congenital alveolar dysplasia [16]. Conditions that only affect proper development of the alveoli were collected under the term growth abnormalities reflecting deficient alveolarization. Examples were infants with trisomy 21, pulmonary hypoplasia, or chronic lung disease of prematurity (BPD). The third group was difficult to appreciate and called “specific conditions of undefined etiology”. It contained two, histological diagnosis pulmonary interstitial glycogenosis and neuroendocrine cell hyperplasia of infancy. The last group contained disorders suggestive of a metabolic abnormality in surfactant metabolism, the surfactant dysfunction disorders. More than 10% of the biopsies were excluded as insufficient and were collected under the label “unclassifiable”. Thereafter, the system was successfully applied to 259 diagnostic biopsies [17].

In 2009 and 2015, this classification system was utilized and expanded to all cases of chILD, irrespective of whether a histopathological diagnosis was available or not and irrespective of the age of the child. To accommodate cases that could not be resolved, additional categories such as unclear respiratory distress in the mature neonate (group Ax), the almost mature neonate (group Ay), and in the non-neonate (group Bx) [18,19] were added. Another important change was the removal of the category “disorders masquerading as ILD” and the introduction of DPLD related to lung vessels’ structural processes (B4) and DPLD related to reactive lymphoid lesions (B5). Due to the large number of 861 chILD cases assessed, a very broad spectrum of ILD was accommodated and some entities were expanded (DPLD in the presumed immunocompromised host or transplanted (B3)) or refocused (DPLD in the presumed immune intact host, related to exposures (B2)). Importantly, this study established and empirically tested a workflow of the pediatric ILD categorization system. The steps included the generation of a final working diagnosis, the decision on the presence or absence of ILD, the decision if the condition assessed was part of a systemic condition or related solely to the lung, and, lastly, the allocation to a predefined disease category and subcategory. Blind re-testing of a random sample set by two independent raters allocated more than 80% of the re-categorized cases identically. Non-identical allocation was due to a lack of appreciation of all available details, insufficient knowledge of the classification rules by the raters, incomplete patient data, and shortcomings of the classification system itself [19].

The French network for pediatric interstitial lung diseases published in 2012 their database categorization system [20]. The categories were linked to the initial ERS scheme from 2004 and accentuated separately granulomatous diseases, metabolic disorders, and infectious ILD.

Such an expansion of the categories was in line with the 2013 published expansion of the chILD classification, based on the histopathological evaluation of 229 biopsies, of which more than 50% were obtained beyond the age of 2 years [21]. The authors added the categories lymphoproliferative disease, small airways disease, interstitial pneumonias unrelated to surfactant protein deficiencies, and other patterns of diffuse lung disease. An important additional notion from this study was the observation that many cases showed mixed histological patterns. Often overlap between groups of disorders more prevalent in infancy were seen. Similarly, patients with a major histological pattern associated with surfactant protein disorders had various minor patterns, including hypoplasia, CPI, DIP, PAP, ELP, NSIP, PH, PIG, or cholesterol pneumonitis.

In 2015, Kitazawa and Kure contributed the Japanese experience. They argued strongly in favor of an etiological classification system [22]. Additionally, in 2015, the team from the children’s interstitial and diffuse lung disease research network applied their chILD classification scheme from 2007 to children with diffuse lung disease and biopsied between the age of 2 to 18 years [23]. They introduced at front a central clinical decision, i.e., was the patient clinically immune compromised or immune competent? This decision may be difficult in everyday practice, as it is often not easily determined [24]. Also in 2015, Armes et al. detailed a pattern-based, algorithmic approach to histological diagnosis of chILD [25]. They stressed and broadened the significance of hypertensive disease as an important component of diffuse lung disease in children. Although several authors have used the 2007 chILD classification system with no or little adaptations [26,27,28], recently it became more and more clear that a new classification of childhood diffuse lung diseases is needed [29].

## 4. Diagnostic Algorithm and Etiologic Grouping of Interstitial Lung Diseases

Usually clinicians suspect an interstitial lung disease if chronic respiratory symptoms, often starting with exercise intolerance and dyspnea, hypoxemia, and diffuse bilateral interstitial imaging abnormalities, are present (see algorithm Figure 2). Other, more common entities are excluded. A broad history and serological investigations are key for initial diagnosis [30]. Advanced testing, in particular, genetic analysis to identify known entities, will support making molecular diagnosis; however, this is currently mainly reserved for a specialized clinical or research setting (see Section 5). Comprehensive consideration of all available data, at best in a multidisciplinary team (MDT), concludes with the most precise diagnosis [31]. This will be the basis for further treatment and judgements of prognosis. Follow up of cases in an active register is highly recommended (Figure 2) [3].

When making the diagnosis as described in the algorithm physicians intuitively try to give a cause for the lung disease, i.e., hypothesize about its etiology. Thus, we group a patient accordingly and have specifiers in mind to describe disease characteristics. The ILD is differentiated into four categories: lung-only (or native parenchymal) disorders, systemic disease-related disorders, exposure-related disorders, and vascular disorders (Table 2).

The native parenchymal disorders are “lung-only” or “isolated pulmonary” conditions, i.e., diseases not part of systemic conditions and not related to exposures or to vascular disorders. An important specifier to group these disorders further is age. Conditions emerging exclusively during the lung development period (pre-natal to 18 years old) and typically manifesting during early childhood are labelled as “developmental” lung disorders (Table 1). The previous grouping into diffuse developmental disorders (A1), alveolarization disorders (A2), infant conditions of undefined etiology (A3), and alveolar surfactant region disorders (A4) is kept (Table 2). Fortunately, due to improved recognition and care, many of these patients will survive into adulthood; thus, pulmonologists need to be aware of conditions such as pulmonary hypoplasia due to diaphragmatic hernia or premature birth.

The “lung-only” conditions may occur at all ages and, nevertheless, are likely to have strong genetic etiology factors, as yet unknown to a large extent. Recently, several conditions previously thought to exclusively occur in childhood, e.g., SFTPC or ABCA3 deficiency, have been demonstrated to occur at all ages [32]. Up to now, in adults these conditions were lumped under interstitial idiopathic pneumonias (Table 1). Future investigations of the biology will help define etiologic entities (see Section 6).

Although having a prominent pulmonary phenotype, many ILDs are systemic disease-related disorders (Table 2). This is a large group and contains well-known conditions such as collagen vascular disorders (CTD-ILD), sarcoidosis, antisynthetase syndrome, or granulomatosis with polyangiitis (GPA, Wegener). Increasing molecularly well-defined entities including aminoacyl-tRNA synthetase deficiencies [33], Birt–Hogg–Dube syndrome, brain-thyroid-lung syndrome, small patella syndrome (TBX4 mutation), telomere-related diseases [34], or ZNFX1 deficiency are differentiated [35]. Patients in this group may be further specified as immunocompetent, immunodeficient, immune dysregulated, auto-inflammatory, or transplanted.

Many ILDs are the result of exposure to inhaled materials. We separated this etiologically important group from the lung-only disorders to put their prevention into the focus. In practice, there will be some overlap; for example, with chronic hypersensitivity pneumonitis the clear demonstration of the relevance of an exposition is often difficult. The earlier such a diagnosis is made, the better the prognosis of the patients will be.

Lastly, vascular disorders, which are often undiagnosed as a lung biopsy may be necessary, represent an important group in the differential diagnosis of ILDs. These conditions only affect the vessels of the lungs. If the vascular disease can be diagnosed as part of a systemic condition, the disease is categorized under “systemic disease-related disorders”. If not, or not yet, the disease is provisionally categorized here and can be specified further with the addition “associated with”, e.g., idiopathic pulmonary hemorrhage with autoimmune features, until a definite diagnosis is made.

Major examples of subcategory entries from the four categories are given in Table 2; the chILD-EU register can make up-to-date lists available (www.childeu.net, accessed on 29 December 2021).

## 5. Genetics Is an Important Step towards Resolving “Idiopathic” Lung Disease

The concept of strong genetic etiologic factors for diffuse lung diseases was developed in pediatrics, as many of the genetically driven disorders manifest early in life. Indeed, the first mono-genetically caused diffuse chronic interstitial lung diseases were identified almost 30 years ago and related to surfactant dysfunction, i.e., caused by mutations in SFTPB, SFTPC, or ABCA3 [36,37,38]. Meanwhile, many other conditions were added, allowing non-invasive and targeted approaches to diagnosis and treatment. Most of the conditions are rare (fewer than 5 in 10,000 people) or ultra-rare. For their study, collection in registers is pivotal, not only to benefit from the accumulated experience in management but also to learn about their natural disease course and underlying mechanisms. There is ongoing exploration of undiagnosed cases from registries, with increasing success rates to identify the etiologic. With technology advances, more and more genetically complex conditions are uncovered [39].

Based on several decades of research into the causes of pediatric and familial ILD in adults and the broad introduction of rare diseases into the public, genetic analysis in patients with fibrosing ILD is primarily done in specialized ILD centers, but it is approaching clinical routine (Figure 2). This may have a profound impact on patient care. It must be clearly stated that there is a current lack of an international consensus statement and recommendations on genetic analysis in ILD. Similar to the numerous other diseases with a genetic base, physicians need expertise and guidance on when to order which genetic test and offer counseling together with the results. In particular, screening of asymptomatic family members is an important field for daily practice, without recommendations yet. Until then, genetic analysis should be primarily done in a well-controlled research setting in specialized ILD centers.

The approximately 10% prevalence of ILD in a first-degree relative of a patient with IPF illustrates the strong familial linkage of ILD [40,41]. In 2016, in adults with suspected mono-genic pulmonary fibrosis, in 40% of the patients a Mendelian disease cause was identified, i.e., a disease-causing genetic variant constellation [42]. Approximately 25% of families with fibrosing lung disease had an identified mutation in genes mostly involved in telomere homeostasis (TERT, TERC, RTEL1, PARN, DKC1, TINF2, NAF1), and less frequently in surfactant homeostasis (SFTPC, SFTPA1, SFTPA2, ABCA3, NKX-2.1). With advancing technology, these numbers continuously increase [43]. The manifestation risks for lung disease caused by the rare genetic variants can be very strong and go far beyond the odds of 20. The latter effect is known from the relatively common constellation of homozygous, less frequent promoter allele rs35705950 of MUC5B encoding mucin-5B [44].

## 6. Next Steps: Systems’ Medicine and Personalized Treatments

Simplified pathophysiological models that explain cell, organ, or body functions may allow easy communication between physicians and with patients [45] but utilize only a fraction of the available information. Currently, large amounts of phenotypic and diverse molecular data are produced [46,47], and efforts are ongoing for their integration to ensure a most comprehensive picture of diseases (Figure 3). In order to integrate the forthcoming wealth of health- and disease-related information, systems’ medicine is in position. These data include knowledge of individual genetic variants, allowing personalized treatments and multi-omics data, such as transcriptomics, proteomics, epigenetics, functional genomics, reverse genetics from transcript manipulation, and deep phenotyping including information from multiple body sensors, environmental exposures, or complex psychosocial influences [48].

## 7. Registers for Diffuse Lung Diseases and Their Working Features

Decent case collections or larger patient cohorts are key for all ongoing research developments in rare diseases. To this end, registers with associated biobanks are pivotal treasures for exploration. To build such cohorts, enthusiastic efforts are necessary for both patients to voluntary participate and physicians to actively submit data and biomaterials. Fortunately, over the last decade many registers for diffuse lung disease and other rare lung diseases were established (see listing in https://clinicaltrials.gov/, accessed on 29 December 2021). For adults, a wealth of national and international registers exists, whereas few do so for pediatric diffuse lung disease. The latter includes the Australasian Registry Network for Orphan Lung Disease (ARNOLD) focusing on chILD in immunocompetent patients, the French register “Respirare”, the United States chILD network, and, in particular, the European chILD-EU register and biobank (www.childeu.net, accessed on 29 December 2021). The last one mentioned offers protocols for diagnosis and initial treatment and a multidisciplinary review of incoming cases, as well as advanced genetic diagnostics and pediatric histopathologic expertise [3]. Registers also serve as very important and efficient tools to assess diagnostic and therapeutic needs of rare disease entities and to prepare for clinical trials therein. Such analyses are of particular interest for collaborations with industry, e.g., to prepare necessary pediatric investigational plans (PIP).

In the context of this paper, an important problem all registers face concerns the disease classification schemes used in their repositories. In particular, when linking the data for common analyses it is necessary to map the diseases and the many descriptors utilized carefully to avoid comparing apples and oranges. For such purposes, ontologies were developed. Ontologies are collections of terms that specify the properties of a complex concept and the relations of the terms. As an example, Ryerson et al. specified a standardized framework to classify the likelihood of a diagnosis for adult fibrotic interstitial lung disease. They defined the terms “confident”, “provisional with high or low likelihood”, and “unclassifiable”. This addition of a clinical modifier and other subontologies, such as clinical course, mode of inheritance, age of manifestation, and frequency, are essential steps towards deep phenotyping of diseases [49]. The Human Phenotype Ontology (HPO) is another example of a collection of terms assisting in describing medically relevant phenotypes so that they are useful for supporting clinical differential diagnosis and translational research interests [50]. These terms allow bio-informatic computations over the clinical phenotype and combining those data with the huge amounts of -omics data for precision medicine. For ILD, we have recently added many novel terms and disease annotations to the current HPO [51].

## 8. The Etiologic Classification of ILD

The chILD-EU register evaluated an etiologic grouping successfully some years ago [19] and continuously rolled it out to more than 1000 cases of diffuse lung diseases. This simple and universal system for diffuse lung disorders separates ILD from airway disorders, pleural disorders, neoplasms, and gross structural abnormalities of the lungs (Figure 1). The four ILD categories defined above (lung-only disorders, exposure-related disorders, systemic disease-related disorders, and vascular disorders) proved robust without the need to generate new groups. This simple, basic structure is easy to recall, and specifiers allow further subcategorizations and descriptions (Table 2). Hierarchical ordering structures are avoided. Modern electronic search systems can accommodate additional complexity from structured term collections.

Sometimes patients with unclassifiable conditions are neglected or lost, as no clear diagnostic label is applied. However, such conditions are the driving force of future developments as they carry an increased likelihood that they harbor extreme phenotypes with potential for strong mono-causal origins. There are numerous examples for new diseases discovered by focusing on such cases [33,35]. We suggest as an important feature including “unclassifiable” conditions into classification systems and indicating why they are inconclusive with the help of modifier terms. In the same line, etiologic disease classifications’ systems can be in place irrespective of if we currently know the exact etiology or not.

Whereas it is globally accepted that children cannot be treated as small adults, and that a closed classification system for pediatric ILD was useful during the establishment of chILD, it is of great advantage to have a common classification system, which accommodates lung diseases from patients of all ages. Important arguments in favor of a common system come from the fact that more and more of the affected children and young adolescents, previously exclusively treated in pediatric pneumology, now reach adulthood [52]. These grown-ups need to be transitioned into medical services for adults. This also concerns knowledge of the neonatal or infancy history and the diseases associated. For example, it is of importance for a pneumologist who cares for a 23-year-old subject to know that the patient was born at 24 weeks of gestation and ventilated for chronic lung disease of prematurity over 4 months. Conversely, there are conditions, such as the telomere-associated lung diseases, that are more frequent in young adults but may also occur in childhood. The pediatric pneumologists should also know such conditions to diagnose them early. Therefore, pediatric specialists in ILD should be included in working groups on ILD classifications or position statements [53]. Inclusion of the pediatric perspective has been realized to some extend by the European Respiratory Society in recent task forces on ILDs.

Pursuing an etiology-based classification has many advantages but is not always easy. In the case of multifactorial etiology, we focus on establishing the most likely etiology in a given patient. If necessary, we fall back to a phenotypic/descriptive diagnosis, judiciously using both concepts, the etiologic one improving clinical research as well as serving the patient in daily clinical practice. Thus, assignment of such cases to any single etiology may be to an extent arbitrary, but in most cases there is a predominant cause to which the case is assigned, which is obviously helpful and valuable for many reasons (Table 3). However, any etiologic classification highly depends on the degree of investigation and investigatory technology. If clinically indicated but not followed by the appropriate investigations, e.g., chest CT, genetic analysis, lung biopsy, or detailed antibody measurements, an advanced diagnosis and classification cannot be made. Any effort should be made for a particular patient to clarify the diagnosis and associated treatment and prognosis. The suggestions made here are not meant to speak for the world but may be a base for discussion towards a comprehensive age-overarching classification of diffuse lung diseases in broad, international, expert consensus groups.

Realistically, no single classification will meet all needs. Thus, ongoing efforts for the integration and merging of equivalent disease concepts are necessary. Different physicians and researchers annotate the relationships and levels of granularity with which they are concerned. Computing technology has to merge equivalent concepts by iterative curator-assisted inference of the different vocabularies. The Mondo Disease Ontology currently does this approach. Mondo, derived from the Latin word ‘mundus’, meaning ‘for the world’, aims to harmonize disease definitions across the world (https://mondo.monarchinitiative.org/, accessed on 29 December 2021).

## 9. Conclusions

We reviewed the development of ILD classifications in pediatric and adult pneumology and suggest an age-overarching etiologic classification of lung diffuse diseases able to accommodate the conditions in an easily memorable way. Genetic testing needs to be integrated more and more into the diagnostic algorithms and may help resolving idiopathic interstitial lung diseases. In particular, the etiologic grouping and longitudinal observation of the diffuse parenchymal lung diseases in registers may support integrating the tremendous -omics data becoming available.

## Figures and Tables

**Figure 1 jcm-11-01747-f001:**
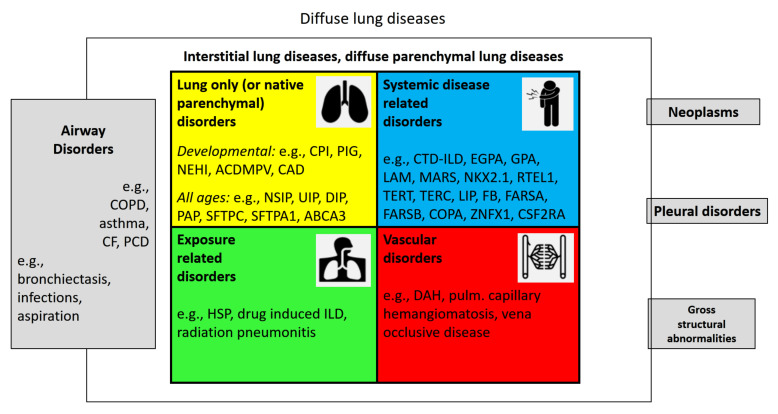
Overview of interstitial (diffuse parenchymal) lung diseases and their neighbors. Diffuse lung diseases involve the entire lungs, i.e., from base to top, peripheral to central, and front to back. Many airway disorders, including COPD, asthma, and cystic fibrosis, can be diffuse, whereas others remain more localized; few pleural diseases, gross structural abnormalities such as congenital pulmonary malformation type 0, or pleural disorders are diffuse. The numerically largest group of diffuse lung diseases are the interstitial (diffuse parenchymal) lung diseases (ILD). These consist of four large etiologic categories of diseases, affecting the lung interstitium or parenchyma, (1) the lung-only (native parenchymal) disorders, (2) the systemic disease-related disorders, (3) exposure-related disorders, and (4) the vascular disorders. ILDs from all age groups are included.

**Figure 2 jcm-11-01747-f002:**
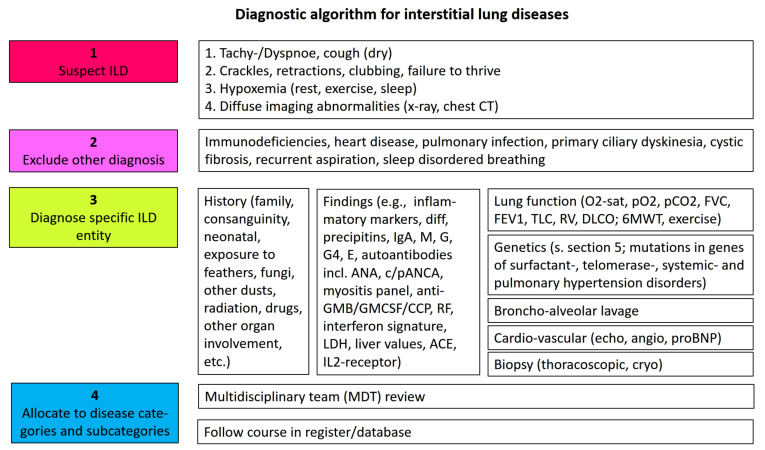
Diagnostic algorithm for the identification and etiologic classification of ILD.

**Figure 3 jcm-11-01747-f003:**
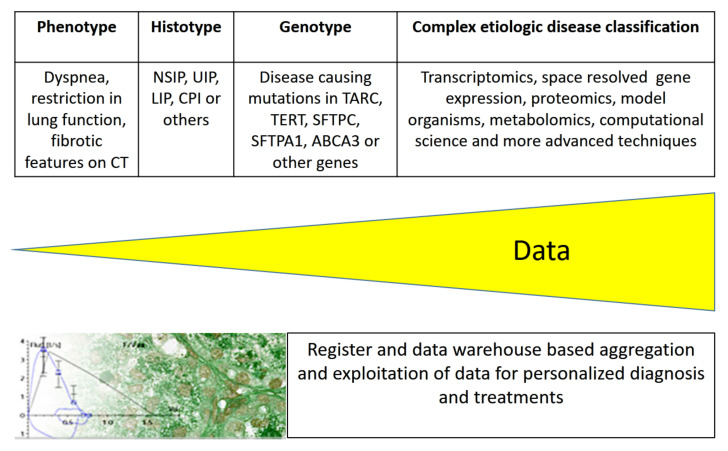
Etiologic disease classification of ILDs drives diagnostic and treatment opportunities. The integration of more and more disease-related data from deeper analysis and systems’ medicine will result in personalized medicine, allowing more precise diagnosis and treatment.

**Table 1 jcm-11-01747-t001:** Compilation of classifications of interstitial lung diseases (ILDs) or diffuse parenchymal lung diseases (DPLDs).

Category (This Paper)	ATS/ERS Internatl. Multi-disciplinary Consensus Classification 2002	ATS/ERS Respiratory Society Statement: Update (Travis et al., 2013)	ERS/ATS Research Statement (Fischer et al., 2015)	ERS Task Force on Chronic Interstitial Lung Disease (Clement 2004)	Diffuse Lung Disease in Children (Deutsch 2007)	Incidence and Classification of Pediatric Diffuse Parenchymal Lung Diseases (Griese et al., 2009)	A National Internet-Linked Database for Pediatric Interstitial Lung Diseases (Nathan et al 2012)	Diffuse Lung Disease in Infancy and Childhood: Expanding the chILD Classification (Rice et al., 2013)	Interstitial Lung Disease in Childhood: Clinical and Genetic Aspects (Kitazawa and Kure 2015)
	adult	adult	adult	pediatric	pediatric	pediatric	pediatric	pediatric	pediatric
Lung only (or native parenchymal) disorders				Congenital disorders	Diffuse developmental disorders	Diffuse developmental disorders (A1)	ILD specific to infancy	Diffuse developmental disorders	Diffuse developmental disorders and growth abnormalities
					Growth abnormalities reflecting deficient alveolarisation	Growth abnormalities deficient alveolarisation (A2)		Growth abnormalities	
					Specific conditions of undefined etiology	Infant conditions of undefined etiology (A3)		Specific conditions of undefined etiology	Conditions specific to infancy
	Idiopathic interstitial pneumonias:			Idiopathic interstitial pneumonias	Surfactant dysfunction disorders	DPLD–related to alveolar surfactant region (A4)	Alveolar disorder related ILD	Histological patterns associated with surfactant disorders	Genetic surfactant dysfunctions
	Idiopathic pulmonary fibrosis (IPF), nonspecific interstitial pneumonia (NSIP), cryptogenic organi-zing pneumonia (COP), acute interstitial pneumo-nia (AIP), desquamative interstitial pneumonia (DIP), idiopathic pleuroparenchymal fibroelastosis			Other forms of interstitial pneumonia		Unclear RDS in the mature/almost mature neonate (Ax/Ay)		Interstitial pneumonia unrelated to surfactant protein disorder	
	Respiratory bronchiolitis-interstitial lung disease (RB-ILD)							Small airways disease	
		Unclassifiable idiopathic interstitial pneumonias				Unclear RDS in the NON-neonate (Bx)	ILD with no diagnosis		
Systemic disease related disorders	DPLD of known cause (e.g. associated with collagen vascular disease)		Interstitial pneumonia with auto-immune features	DPLD of known association (e.g. connective tissue disorders)	Disorders related to systemic disease processes	DPLD—related to systemic disease processes (B1)	Systemic disease associated ILD	Disorders related to systemic disease	Interstitial lung disease related to a primary systemic disease
	Granulomatous DPLD e.g. sarcoidosis						Granulomatous diseases		
	Other forms of DPLD (e.g. LAM, HX)						Metabolic disorders		
					Disorders of the immunocompromised host	DPLD—In the immunocompro-mised host or transplanted (B3)	Infectious ILD	Host disorders (presumed immuno-compromised)	
	Lymphoid interstitial pneumonia (LIP)	Idiopathic lymphoid interstitial pneumonia				DPLD—Related to reactive lymphoid lesions (B5)		Lymphoproliferative disease	
Exposure-related disorders	DPLD of known cause e.g. drugs			DPLD of known association (e.g. drug, aspiration, infection, environment)	Disorders of the normal host presumed immune intact	DPLD—In the presumed immune intact host, related to exposures (infect/non-infect) (B2)	Exposure related ILD Hypersensitivity pneumonitis	Host disorders (presumed immune intact)	Exposure-related Interstitial lung disease
Vascular disorders					Disorders masquerading as interstitial disease	DPLD—Related to lung vessels structural processes (B4)	Alveolar vascular disorder related ILD	Disorders masquerading as interstitial disease	

**Table 2 jcm-11-01747-t002:** Classification of interstitial lung diseases (ILDs) used in the chILD-EU register (full list available www.childeu.net, accessed on 29 December 2021).

Category	Specifier	Subcategories (Major Examples)	
Lung only (native parenchymal) disorders	Developmental	Diffuse (A1) - Alveolar capillary dysplasia, misalign pulmonary veins (ACDMPV) - Congenital alveolar dysplasia (CAD)	Alveolarization (A2) - Pulmonary hypoplasia - Pulmonary hypoplasia associated with diaphragmatic hernia - Chronic lung disease of prematurity (BPD-cLD)
		Infant tachypnea (A3) - Chronic tachypnea of infancy (CTI) (usual, aberrant) - Neuroendocrine cell hyperplasia of infancy (NEHI) - Pulmonary interstitial glycogenosis (PIG)	Surfactant region (A4) - Chronic pneumonitis of infancy (CPI) - Surfactant protein B deficiency - Other surfactant dysfunction disorders - Undefined ILD in mature neonate - Undefined ILD in almost (gest. age 30–36 weeks) mature neonate
	All ages	- ABCA3 deficiency - Surfactant protein C deficiency - Lipoid pneumonitis, cholesterol pneumonia - Nonspecific interstitial pneumonia (NSIP) - Idiopathic pulmonary fibrosis (IPF)/Usual interstitial pneumonitis (UIP) - Cryptogenic organizing pneumonia (COP)	- Acute interstitial pneumonia (AIP) - Desquamative interstitial pneumonia (DIP) - Idiopathic pleuro-parenchymal fibroelastosis - Eosinophilic pneumonitis - Undefined ILD in NON-neonate - Unclassifiable idiopathic interstitial pneumonias
Systemic disease related disorders	Immuno competent	- Collagen vascular/connective tissue related disorders (CTD-ILD) - Eosinophilic granulomatosis with polyangiitis (EGPA, Churg Strauss) - Granulomatosis with polyangiitis (GPA, Wegener) - Metabolic disorders - Disorders related to chromosomal abnormalities - Aminoacyl-tRNA synthetase (-ARS) deficiency - Birt-Hogg-Dube syndrome - Granulomatous diseases	- Diffuse alveolar hemorrhage due to vasculitic disorders - ITGA3 (Integrin α3) mutations with kidney, lung, skin disease - Niemann-Pick disease - Nkx2.1 gene defect (brain-thyroid-lung syndrome) - Lysinuric proteinuria - PAP secondary to associated diseases - Sarcoidosis - TBX4 mutation (small patella syndrome) - Hoyeral Hreidasson Syndrome (congenital dyskeratosis) - Telomere-related-diseases
	Immuno deficient	- Phagocyte-PAP.CSF2RA or CSF2RB (PAP due to GMCSF receptor deficiency)	- Phagocyte-GATA2 def (MonoMac syndrome).GATA2
	Transplanted	- Chronic allograft dysfunction, idiopathic pneumonia syndrome	
	Immune- dysregulated	- Autoimmune pulmonary alveolar proteinosis (PAP) - Antisynthetase syndrome - Celiac disease/pulmonary hemorrhage (Lane-Hamilton) - Anti-basement membrane antibody disease (Goodpasture) - Granulomatous lymphocytic interstitial lung disease (GLILD)	- Follicular bronchitis/ bronchiolitis - Lymphocytic interstitial pneumonia (LIP) - Interstitial pneumonia with auto-immune features (IPAF) - ZNFX1 deficiency - Immune-dysregulated Hermansky Pudlak syndrome
	Auto-inflammatory	- COPA defect - STING-associated vasculopathy, infantile-onset.TMEM173	- OAS1 deficiency - Other interferonopathies
Exposure related disorders	Non-infectious	Exogen allergic alveolitis/hypersensitivity pneumonitis - Drug induced ILD - Pneumoconiosis	- Radiation lung injury - Respiratory bronchiolitis-interstitial lung disease (RB-ILD) - Occupational lung diseases
	Infectious	- Infectious/post-infectious bronchiolitis obliterans (PIBO) - Diffuse panbronchiolitis	- Mac-Leod-Swyer-James-Syndrome
Vascular disorders		- Diffuse pulmonary (alveolar) hemorrhage (DPH) - Idiopathic pulmonary hemorrhage (hemosiderosis) (IPH) with or without associated features - Pulmonary capillaritis	- Pulmonary capillary hemangiomatosis (PCH) - Pulmonary vena-occlusive disease - Lymphangiomatosis

**Table 3 jcm-11-01747-t003:** Advantages of an etiology-based classification of interstitial lung diseases.

Etiology is the driving force for most progress towards treatment of conditions
Etiology, as the one major criterion, is primarily used for all classification-decisions in the system
Hierarchically simple and universal system, which can be easily memorized on a major level
Flat, two-level structure (four major, primary levels and not branched second level, which can accommodate any large number of different entities)
Minimizes contradictions often experienced in hierarchical classifications
Open to further subclassification by convenient specifiers and descriptors
Avoids neglect of unclassifiable conditions by also including conditions with currently not exactly known etiology or even for cases in which the absence of information includes classes as “undefined” or “probable”
Ability to annotate relationships between etiologies at all levels of granularity
The system does not ignore tradition of old classification schemes
Open to all ages, accommodating easily age-specific disease manifestations
Supports closer collaboration of pediatric and adult pneumologists, and, for the patients, an improved transition of grown-ups into adult medical services
Computational methods may overcome the relatively large number of conditions resulting, which is seen by some as a disadvantage
Computations over phenotype, multi-omics, environmental, or psychosocial data easily possible
Proven applicability in electronic databases of a large ILD register
Usage of standardized collections of terms (ontologies) for merging with other machine-based systems (big data analysis)

## Data Availability

An updated classification of interstitial lung diseases (ILDs) used in the chILD-EU register is available from www.childeu.net (accessed on 29 December 2021).

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
