# Peer review of "Etiologic Classification of Diffuse Parenchymal (Interstitial) Lung Diseases"

_jcm, 2022, doi:10.3390/jcm11061747_

Round 1
Reviewer 1 Report
Given the nature of this submission as a perspective, I believe the content to be appropriate and a useful perspective for readers of this journal from a content expert. I read with great interest the discussion of the history of lumping and splitting in ILD and have only a few minimal considerations for the author that would not preclude publication from my point of view.
In Figure 2. While clearly the algorithm is not meant to be exhaustive, the choice of serologies in the evaluation of CTD-ILD is puzzling. For instance, the ATS recommends (in adults) that in cases where IPF is under consideration that the serologies should at a minimum include ANA, anti-CCP, RF and consider screening for myositis with muscle enzymes +/- myositis panel. While for less clear cases such as NSIP or indeterminate UIP pattern radiographs it is very reasonable to broadly expand the serologies, I would consider at least including the guideline-directed minimum serologies as described. Also I have concerns about including "genetics' in the diagnostic algorithm section where these tests are not readily available in US clinical practice and have (as of yet) no clear bearing on treatment decisions, I think at least some mention that of that caveat might be useful.
Author Response
Reviewer 1
I thank the reviewer for appreciating the manuscript and the valuable comments.
C1 Given the nature of this submission as a perspective, I believe the content to be appropriate and a useful perspective for readers of this journal from a content expert. I read with great interest the discussion of the history of lumping and splitting in ILD and have only a few minimal considerations for the author that would not preclude publication from my point of view.
R1 Thank you very much
C2 In Figure 2. While clearly the algorithm is not meant to be exhaustive, the choice of serologies in the evaluation of CTD-ILD is puzzling. For instance, the ATS recommends (in adults) that in cases where IPF is under consideration that the serologies should at a minimum include ANA, anti-CCP, RF and consider screening for myositis with muscle enzymes +/- myositis panel. While for less clear cases such as NSIP or indeterminate UIP pattern radiographs it is very reasonable to broadly expand the serologies, I would consider at least including the guideline-directed minimum serologies as described.
R2 Changed as suggested and also included latest guideline in the manuscript.
C3 Also I have concerns about including "genetics' in the diagnostic algorithm section where these tests are not readily available in US clinical practice and have (as of yet) no clear bearing on treatment decisions, I think at least some mention that of that caveat might be useful.
R3 Yes, thank you; I included a paragraph in section 5. on the current lack of international consensus recommendations for genetics, the need to offer counselling and the primary usage in specialized ILD centers. Also, the figure now refers to the text in section 5.
Reviewer 2 Report
The manuscript is well written but a detailed classification DPLD already exists as given by the ATS/ERS Guidelines and so a more elaborate classification with respect to age should be defined on your part specifically defining the age cut-offs for various ILDs.
Author Response
Reviewer 2
I thank the reviewer for appreciating the manuscript and the valuable comments.
C1 The manuscript is well written but a detailed classification DPLD already exists as given by the ATS/ERS Guidelines and so a more elaborate classification with respect to age should be defined on your part specifically defining the age cut-offs for various ILDs.
R1 Knowledge on age-cut offs of currently defined ILD´s is limited; in particular due to advances in care and genetic diagnosis age ranges blur more and more; diseases previously only prevalent in infants can now be identified in young and older adults. This is discussed in section 8. of the manuscript.
Reviewer 3 Report
General comments
- The author has tackled a very complex and confusing subject and correctly provides a sensible argument for developing a universal classification system for ILD that spans all ages. The author’s insight on the subject had grasping of historical aspects is obvious. However, this reviewer was a bit lost with the detailed descriptions of the history of ILD classifications in children and adults and questions the relevance and importance to readers. Table 1 provides a good overview of the various classifications in historical context and in my view is sufficient for the most readers. I would suggest revising these sections and manuscript with a more concise/simplified historical overview that outlines the flaws with past/current classifications , and more emphasis and focused discussion on the proposed new classification and why it is needed.
- The relevance of the proposed aetiological classification system and how it relates to future “precision” treatment or why this this classification is better than others is not clear. For example, with Lung Only category , there is a wide range of different aetiologies of ILD, some with genetic origins others not. Some potentially treatable, others not. Some with genetic aetiology , others not. More clarity on this would be helpful. Lungs Only seems more an anatomical distribution classification.
- As the author has stated, the spectrum of ILDs in adults that originate in childhood is proportionately increasingly s children survival longer or a genetic aetiology is identified in ILDs previously labelled as “idiopathic. Does the author believe there will be buy-in for ta new “universal” ILD classification by adult physicians /researchers in this field?
Specific comments
- Paragraph line 80-88: this paragraph seems out of place and does not follow the previous page- please check.
- Table 2: the formatting is not clear which makes the table difficult to read. What do letters in the column “ previous labels” mean?
Figure 3: the meaning and purpose of this figure is not clear and seems disconnected from the title: “ Etiologic disease classification of ILDs drives diagnostic and treatment opportunities
Author Response
Reviewer 3
I thank the reviewer for appreciating the manuscript and the valuable comments.
General comments
C1 The author has tackled a very complex and confusing subject and correctly provides a sensible argument for developing a universal classification system for ILD that spans all ages. The author’s insight on the subject had grasping of historical aspects is obvious. However, this reviewer was a bit lost with the detailed descriptions of the history of ILD classifications in children and adults and questions the relevance and importance to readers. Table 1 provides a good overview of the various classifications in historical context and in my view is sufficient for the most readers. I would suggest revising these sections and manuscript with a more concise/simplified historical overview that outlines the flaws with past/current classifications , and more emphasis and focused discussion on the proposed new classification and why it is needed.
R1 Thank you for this supporting appraisal; as another reviewer read with great interest this section on the history of ILD, I reviewed the section, cut out an overemphasized part and left the rest unchanged.
C2 The relevance of the proposed aetiological classification system and how it relates to future “precision” treatment or why this this classification is better than others is not clear. For example, with Lung Only category , there is a wide range of different aetiologies of ILD, some with genetic origins others not. Some potentially treatable, others not. Some with genetic aetiology , others not. More clarity on this would be helpful. Lungs Only seems more an anatomical distribution classification.
R2 Thank you for this important point which I clarified in the text by introducing a new paragraph on that a a table 3, listing the advantages of an etiology based classification.
C3 As the author has stated, the spectrum of ILDs in adults that originate in childhood is proportionately increasingly s children survival longer or a genetic aetiology is identified in ILDs previously labelled as “idiopathic. Does the author believe there will be buy-in for ta new “universal” ILD classification by adult physicians /researchers in this field?
R3 This is an interesting question; at minimum I hope that the suggestions made in the manuscript may be a base for discussion towards a comprehensive age-overarching classification of diffuse lung diseases in broad, international expert consensus groups. This is now also expressed in the text on page 15.
Specific comments
C4 Paragraph line 80-88: this paragraph seems out of place and does not follow the previous page- please check.
R4 This was the figure legend and difficult to differentiate by the same font, was determined by the complex format of MDPI manuscripts. Used now italic for figure legends.
C5 Table 2: the formatting is not clear which makes the table difficult to read. What do letters in the column “ previous labels” mean?
R5 Good advice; I removed that column from the table and integrated only the previously used lables in brackets under the section “Development” and explain this in the text.
C6 Figure 3: the meaning and purpose of this figure is not clear and seems disconnected from the title: “ Etiologic disease classification of ILDs drives diagnostic and treatment opportunities
R6 Thank you; yes, figure was wrongly moved by copying, now under section 6, as intended.